# Self-Reconstructed Metal–Organic Framework-Based Hybrid Electrocatalysts for Efficient Oxygen Evolution

**DOI:** 10.3390/nano14141168

**Published:** 2024-07-09

**Authors:** Kunting Cai, Weibin Chen, Yinji Wan, Hsingkai Chu, Xiao Hai, Ruqiang Zou

**Affiliations:** 1Beijing Key Laboratory for Theory and Technology of Advanced Battery Materials, School of Materials Science and Engineering, Peking University, No. 5 Yiheyuan Road, Haidian District, Beijing 100871, China; 2State Key Laboratory of Heavy Oil Processing, China University of Petroleum, No. 18 Fuxue Road, Changping District, Beijing 102249, China

**Keywords:** metal–organic frameworks, structure reconstruction, oxygen evolution reaction, hybrid electrocatalysts

## Abstract

Refining synthesis strategies for metal–organic framework (MOF)-based catalysts to improve their performance and stability in an oxygen evolution reaction (OER) is a big challenge. In this study, a series of nanostructured electrocatalysts were synthesized through a solvothermal method by growing MOFs and metal–triazolates (METs) on nickel foam (NF) substrates (named MET-M/NF, M = Fe, Co, Cu), and these electrocatalysts could be used directly as OER self-supporting electrodes. Among these electrocatalysts, MET-Fe/NF exhibited the best OER performance, requiring only an overpotential of 122 mV at a current density of 10 mA cm^−2^ and showing remarkable stability over 15 h. The experimental results uncovered that MET-Fe/NF underwent an in situ structural reconstruction, resulting in the formation of numerous iron/nickel (oxy)hydroxides with high OER activity. Furthermore, in a two-electrode water-splitting setup, MET-Fe/NF only required 1.463 V to achieve a current density of 10 mA cm^−2^. Highlighting its potential for practical applications. This work provides insight into the design and development of efficient MOF-based OER catalysts.

## 1. Introduction

The rapid consumption of fossil fuels has led to a series of energy and environmental problems, prompting researchers to develop sustainable and clean energy technologies [1,2,3]. The electrocatalytic oxygen evolution reaction (OER) plays a critical role in renewable energy conversion and storage technologies, such as metal–air batteries, water splitting, and fuel cells [4,5,6]. However, the OER process involves a complex four-electron transfer pathway, and its slow kinetics significantly limit the efficiency of energy conversion systems [7,8]. Catalysts can efficiently reduce the overpotential and improve the OER process, so the proper design and development of catalysts are important. Currently, iridium dioxide (IrO_2_) and ruthenium dioxide (RuO_2_) are considered efficient OER catalysts, but their scarcity, high cost, and low stability under alkaline conditions severely hinder their industrial applications [9,10,11]. Therefore, a key goal for researchers is to design and develop a cost-effective, highly active, and durable non-precious metal OER electrocatalyst [12,13,14,15].

In recent years, metal–organic frameworks (MOFs), which are synthesized through the periodic assembly of organic ligands and metal ions/clusters, have garnered widespread application in the field of electrocatalysis due to their easily tunable structures and functionalities [16,17,18,19,20,21,22,23,24]. Nevertheless, the limited conductivity and chemical stability of many MOFs pose significant challenges to their direct use in electrocatalytic processes [25,26,27]. Metal–triazolate (MET), [M(C_2_N_3_H_2_)_2_]*_n_*, is a new family of MOFs that could be prepared by combining 1H-1,2,3-triazole and divalent metal ions (Fe, Cu, Co, and so on) to give six isostructural metal–triazolates (termed MET-M, M = Fe, Cu, Co, and so on). MET frameworks have permanent porosity and display surface areas as high as some of the most porous zeolites, with one member of this family, MET-Fe, exhibiting significant electrical conductivity [28]. Notably, the in situ growth of MOFs on a conductive substrate can be regarded as a facile strategy to construct self-supported electrodes, which can considerably accelerate electron and mass transfer. In addition, 3D metal foams, such as nickel foam (NF), have previously been reported as a well-known class of electrocatalytic support materials for MOFs [29,30]. Significantly, the transformation of many MOF materials into metal (oxy)hydroxides during the alkaline OER process is a widely observed phenomenon [31,32,33,34], suggesting that these newly formed metal hydroxides could serve as catalytically active sites for OER, such as iron oxyhydroxides (FeOOH), nickel oxyhydroxides (NiOOH), and nickel iron-layered double hydroxides (NiFe-LDH) [19,35,36]. Furthermore, single-component catalysts are often limited in their ability to improve performance, suggesting that the design of composite catalysts with multiple active elements may be a more effective approach to enhance overall catalytic efficiency due to a catalytic synergistic effect [37,38,39,40,41,42]. The interaction between different active sites in such catalysts can refine and accelerate reaction processes, thereby enhancing the catalytic activity [43,44,45].

Considering the above context, in this study, we have successfully synthesized metal–triazolates, (MET-M, [M(C_2_N_3_H_2_)_2_]*_n_*, M = Fe, Cu, Co), on nickel foam via a solvothermal method, which can directly serve as self-supporting OER electrodes. The best sample MET-Fe/NF material undergoes in situ restructuring during the OER process, and the resulting mixed catalyst with multiple active components (FeOOH, NiOOH, and NiFe-LDH) exhibits excellent OER electrocatalytic activity. The MET-Fe/NF material achieves a current density of 10 mA cm^−2^ with an overpotential of only 122 mV and a small Tafel slope of 34.5 mV dec^−1^, outperforming commercial RuO_2_, and it maintains good stability over 15 h. In two-electrode water splitting applications, the energy conversion device equipped with this catalyst operates at a low voltage of 1.463 V to continuously produce hydrogen and oxygen at a current density of 10 mA cm^−2^, demonstrating the promising practical application prospects of the catalyst. This work offers an effective strategy for the design of highly efficient MOF-based electrocatalysts.

## 2. Experimental

### 2.1. Materials

In this study, all chemicals and reagents were of analytical grade and used without further purification. FeCl_2_ (purity: 99%) was purchased from Konoscience Reagent Co., Ltd. (Shanghai, China). CoCl_2_ (purity: 99%) was purchased from Macklin Reagent Co., Ltd. (Seattle, WA, USA). Cu(NO_3_)_2_.*x*H_2_O (purity: 99%) was purchased from Beijing Tongguang Fine Chemicals Co., Ltd. (Beijing, China). 1H-1,2,3-triazole (purity: 98%) was purchased from Shanghai D&B Co., Ltd. (Shanghai, China). N,N′-dimethylformamide (DMF) (purity: 99.8%) was purchased from Concord Co., Ltd. (Singapore). Commercial NF (purity: 99%) (thickness: 0.5 mm) was provided by Kunshan Guangjiayuan Co., Ltd. (Kunshan, China).

### 2.2. Preparation of MET-Fe/NF

First, FeCl_2_ (8.5 mmol) was dissolved in 50 mL of a DMF solution. The solution was then transferred to a Teflon reactor. After that, 1H-1,2,3-triazole (25.5 mmol) was added to the solution and the NF substrate was immersed in the solution. The samples were then obtained after being placed in an oven at 120 °C for 48 h. Finally, after cooling down to RT, the resulting MET-Fe/NF was washed three times with alcohol and dried at 80 °C for 8 h.

### 2.3. Preparation of MET-Co/NF and MET-Cu/NF

The preparation procedures of MET-Co/NF and MET-Cu/NF were similar to that of MET-Fe/NF, except that the metal sources were CoCl_2_ and Cu(NO_3_)_2_.*x*H_2_O, respectively.

### 2.4. Preparation of RuO_2_/NF and Pt/C/NF

For comparison, the commercial RuO_2_ dispersion was prepared using ultrasonication with 4 mg of RuO_2_, 500 μL ethanol, and 100 μL 5 wt% Nafion. After homogenization, 30 μL of the dispersion was drip-coated on NF (1 cm × 1 cm) at a loading of 0.2 mg cm^−2^. The preparation procedure of Pt/C/NF was similar to that of RuO_2_/NF, except that the metal source was commercial Pt/C.

### 2.5. Characterization

The crystallographic structures of the materials were determined using a Rigaku SmartLab 9 kW diffractometer (Tokyo, Japan) with Cu Kα radiation (λ = 1.5406 Å). Nitrogen sorption isotherms were measured at 77 K on a Quantachrome Autosorb-IQ gas adsorption analyzer (Boynton Beach, FL, USA). From the adsorption isotherms, the specific surface areas were calculated using the Brunauer–Emmett–Teller (BET) method. The pore size distributions were obtained using the quenched solid density functional theory (QSDFT) method. The microstructure and morphology were examined by using a scanning electron microscope (SEM; Hitachi S-4800 microscope, Tokyo, Japan). The surface characterization of elemental electronic states was measured by the Krayos AXIS Ultra DLD electronic energy spectrometer (Shimadzu, Tokyo, Japan) (XPS; Axis Ultra imaging photoelectron spectrometer with the monochromatic Al Kα line). The mid-infrared absorptive spectrum of samples was determined using an ATR-FTIR spectrometer (Bruker, VERTEX 80V, Berlin, Germany).

### 2.6. Electrochemical Measurements

The electrocatalytic properties of OER were measured using a three-electrode system on a CHI660 workstation (Chenhua, Shanghai, China), in which NF-based electrodes, Hg/HgO electrodes, and platinum electrodes served as the working, reference, and counter electrodes in a 1 M KOH solution, respectively. The potentials were referred to the reversible hydrogen electrode (RHE) according to the following equation [46]: E_RHE_ = E_Hg/HgO_ + 0.098 V + 0.059 × pH. The linear sweep voltammetry (LSV) polarization curves of the electrodes were tested at a scan rate of 5 mV s^−1^ with 95% iR compensation. The Tafel slopes were derived from polarization curves. CV tests were performed at sweep rates ranging from 20 to 100 mV s^–1^ in a non-Faradaic region (0.912~1.012 V vs. RHE), then double-layer capacitance (C_dl_) was calculated using the CV results. Electrochemical impedance spectroscopy (EIS) was performed in the frequency range from 100 kHz to 0.1 Hz at open circuit potential. Chronopotentiometry measurement was performed at a current density of 10 mA cm^−2^ without iR compensation. Electrochemical water splitting was measured using a two-electrode system.

## 3. Results and Discussion

The synthesis method for the materials is illustrated in Figure 1, based on adaptations from the literature references [28]. The process involved utilizing Fe^2+^ ions as the metal source, 1H-1,2,3-triazole as the organic ligand, and DMF as the solvent, with nickel foam serving as the growth substrate for the composite material. The procedure involved a one-pot solvothermal reaction in a reaction vessel at 120 °C for 48 h, culminating in the production of MET-Fe/NF sample material. The chosen organic ligand, 1H-1,2,3-triazole, is characterized by a five-membered ring structure containing three nitrogen atoms, which are differentiated into two types, N1 and N2, based on subtle differences in their chemical environment (Appendix A). During the reaction, all three nitrogen atoms are involved in coordination with the metal atoms, with each metal center engaging in octahedral coordination with six nitrogen atoms, thereby constructing a three-dimensional MOF framework endowed with a porous architecture (Appendix A). By changing the metal ions to Co^2+^ or Cu^2+^, two additional comparative samples, MET-Co/NF and MET-Cu/NF, were synthesized. In contrast to the bare nickel foam (Appendix A), the surfaces of all synthesized samples were uniformly coated with a layer of powdery substance, with the MET-Fe/NF sample exhibiting a black-brown powder (Appendix A), the MET-Co/NF sample showing a yellow-brown powder (Appendix A), and the MET-Cu/NF sample displaying a blue powder (Appendix A). These initial observations confirm the feasibility and universal applicability of synthesizing MOFs/NF composites through a solvothermal one-pot method. The array of MOFs/NF composite materials thus obtained can be directly used as self-supporting electrocatalytic electrodes (with an effective electrode area of 1 cm^2^ square), streamlining the subsequent testing of OER performance as well as the analytical characterization of the catalytic materials.

To further determine the phase composition of the samples, X-ray diffraction (XRD) characterization was performed on the samples obtained. As depicted in Figure 2a, the XRD patterns revealed strong peaks corresponding to the nickel foam substrate (PDF# 70-0989), indicating the origin of the Ni peaks. Analysis of the XRD patterns within the 5° to 40° range revealed peaks that were consistent with those in the simulated MET-Fe XRD spectrum, suggesting the successful growth of MET-Fe powder on the surface of the nickel foam. XRD characterization of the precipitate obtained from the one-pot solvothermal reaction, which is homologous to the material on the MET-Fe/NF surface (Appendix A), further confirmed that the powder on the MET-Fe/NF sample surface is indeed MET-Fe. Phase analysis was also conducted on the comparative samples MET-Co/NF and MET-Cu/NF. Although the XRD peak signals from the nickel foam substrate were dominant (Appendix A), additional XRD tests on the homologous solids formed during the reaction confirmed that the powders on the surfaces of the comparative samples were MET-Co (Appendix A) and MET-Cu (Appendix A), respectively. Furthermore, Brunauer–Emmett–Teller (BET) surface area analysis revealed that the MET-Fe powder on the MET-Fe/NF sample surface possesses a high specific surface area (up to 446 m^2^/g) and features a plethora of micropores and mesopores (Figure 2b), which would facilitate extensive contact between the MET-Fe/NF sample and the electrolyte.

Scanning electron microscopy (SEM) was used to investigate the morphology and elemental composition of the samples. Appendix A shows the nickel foam substrate as a metallic framework with pores on the order of 100 μm. The elemental mapping presented in Appendix A shows a uniform distribution of oxygen elements over the nickel foam surface. Figure 3a shows that the nickel foam substrate of the MET-Fe/NF sample is thoroughly coated with a solid powder layer, while higher magnification images (Figure 3b) show that this layer is composed of densely packed nano-sized MOF particles. Figure 3c provides a detailed examination of the MET-Fe growth on the nickel foam surface, while the uniform distribution of Ni, O, Fe, and N elements in Figure 3d confirms the uniform coverage of MET-Fe over the nickel foam. SEM characterization was also performed on the MET-Co/NF and MET-Cu/NF samples. Appendix A shows that there are relatively few solids grown on the surface of MET-Co/NF. In contrast, Appendix A shows that the bulk solids in MET-Cu/NF fill most of the space within the nickel foam substrate. The different morphological composite structures of the samples may influence the subsequent catalytic processes.

The OER performance of MET-Fe/NF and reference samples was initially assessed using a standard three-electrode system with 1M KOH as the electrolyte. The linear sweep voltammetry (LSV) curves depicted in Figure 4a indicate that MET-Fe/NF exhibits superior OER performance. As shown in Figure 4b, MET-Fe/NF requires only a 122 mV overpotential to achieve a current density of 10 mA cm^−2^, outperforming MET-Co/NF (385 mV), MET-Cu/NF (476 mV), bare nickel foam (NF) (375 mV), and the commercial RuO_2_/NF catalyst (354 mV). The relatively lower OER performance of MET-Co/NF and MET-Cu/NF compared to MET-Fe/NF can be attributed to three main reasons. First, the Fe element is likely to offer unique advantages in the nickel foam OER catalyst system due to its inherent electronic properties and catalytic efficiency [47,48,49,50]. Second, MET-Fe is characterized by its inherent electrical conductivity with a value of 0.77 × 10^−4^ S cm^−1^ reported by Yaghi et al. [28], which is essential for catalysis. The projected density of states (PDOS) calculation results (Appendix A) of MET-Fe indicate that MET-Fe seems to be a semiconductor (conductivities in the range from 10^−8^ to 10^2^ S cm^−1^) considering its conductivity value. However, MET-Co and MET-Cu exhibit insulating properties like most other MOFs. This difference in conductivity could adversely affect the electron mobility within the composite, especially when compared to the baseline performance of pure nickel foam. Thirdly, the interaction between the MOFs and the nickel foam substrate shows minimal surface growth for MET-Co and excessive coverage by MET-Cu, which could clog the nickel foam structure and affect its effectiveness. In contrast, MET-Fe forms a uniform and dense growth layer on the nickel foam surface, ensuring the full participation of MET-Fe as the active material without affecting the conductivity of the substrate and its interaction with the electrolyte. These results highlight not only the superior performance of MET-Fe/NF over commercial RuO_2_ but also its impressive material uniqueness within the MET/NF series of composites.

Figure 4c reveals that the Tafel slope of MET-Fe/NF is 34.5 mV dec^−1^, which is significantly lower compared to MET-Co/NF (204.7 mV dec^−1^), MET-Cu/NF (183.4 mV dec^−1^), bare nickel foam (NF) (91.2 mV dec^−1^), and RuO_2_/NF (121.1 mV dec^−1^), highlighting the superior reaction kinetics of MET-Fe/NF. The electrochemical surface area (ECSA) serves as a key indicator of catalytic efficiency, closely linked to the double layer capacitance (C_dl_). Cdl values were derived from the analysis of cyclic voltammetry (CV) curves across various scan rates within the non-Faradaic region, as shown in Appendix A. Illustrated in Figure 4d, the calculated C_dl_ values for NF, MET-Co/NF, MET-Cu/NF, and MET-Fe/NF are 1.68 mF cm^−2^, 1.66 mF cm^−2^, 1.89 mF cm^−2^, and 8.19 mF cm^−2^, respectively, with MET-Fe/NF exhibiting the highest C_dl_ value, indicating the highest number of active catalytic sites during the reaction.

The turnover frequency (TOF) is another important parameter to evaluate the electrocatalysts’ activity [51]. The parameter was calculated with TOF = jA/4Fm, where j is current density, A is the surface area of the electrode, 4 indicates the number of electrons consumed for O_2_ evolution from water, F is the Faraday constant, and m is the number of active species [52]. Appendix A shows TOF curves at various overpotentials, indicating that the TOF values of all catalysts exhibited a monotonic increase with increasing overpotential. Among all other catalysts, the MET-Fe/NF exhibits the highest intrinsic catalytic activity at the same overpotential, suggesting the best OER performance.

Electrochemical impedance spectroscopy (EIS) facilitates the understanding of charge transfer at the reaction interface and provides Nyquist plots (Appendix A). The charge transfer resistance (R_ct_) values derived from the Nyquist plots reveal that MET-Fe/NF has the smallest R_ct_ value of only 0.36 Ω, significantly lower than those of MET-Co/NF (3.97 Ω), MET-Cu/NF (3.86 Ω), NF (2.23 Ω), and RuO_2_/NF (2.08 Ω). This suggests that MET-Fe/NF possesses superior charge transfer capabilities, which is conducive to improving the OER process. These results collectively demonstrate the exceptional OER catalytic activity of MET-Fe/NF compared to the other samples. Stability tests were also carried out on MET-Fe/NF at a current density of 10 mA cm^−2^. The results indicate that MET-Fe/NF exhibits a mere performance degradation after 15 h of reaction, as reflected by the little change (+9 mV) in voltage values, showcasing remarkable stability (Figure 4e). A comparison of MET-Fe/NF with recently reported MOF/metal foam-based OER catalysts reveals superior performance in terms of both overpotential and Tafel slope [53,54,55,56,57,58,59,60,61] (Figure 4f). Taken together, these results establish MET-Fe/NF as an advanced OER catalyst that combines high catalytic activity with remarkable stability.

To assess the practical application performance of MET-Fe/NF further, it was combined with Pt/C/NF in a two-electrode system for a water-splitting experiment conducted in 1 M KOH electrolyte. The LSV curve of the MET-Fe/NF||Pt/C/NF two-electrode electrolysis setup, shown in Figure 5a, demonstrates that this water-splitting apparatus needs just a low cell voltage of 1.463 V to reach an electrolysis current density of 10 mA cm^−2^, highlighting its remarkable water-splitting efficiency and superiority to the commercial RuO_2_-based two-electrode system (Appendix A). Insets in the figure illustrate the formation of gas bubbles on the electrode surfaces during the reaction, which correspond to oxygen and hydrogen gases generated by the anodic OER and the cathodic hydrogen evolution reaction (HER), respectively. The stability test results, depicted in Figure 5b, indicate that after 15 h of continuous operation at 10 mA cm^−2^, splitting water into oxygen and hydrogen, the voltage required to drive the energy conversion device increased only by 20 mV, maintaining stable performance over a long period of time. These results highlight the efficient energy conversion efficiency and remarkable stability of MET-Fe/NF in water-splitting devices, demonstrating its broad application prospects in the large-scale production of hydrogen and oxygen gases.

To investigate the composition of the high catalytic activity sites in MET-Fe/NF for the OER, we characterized the sample after a prolonged series of tests using SEM, XRD, and XPS. The SEM images in Figure 6a reveal morphological changes in MET-Fe/NF after testing compared to its initial state, with several regions retaining their original morphology while others display a noticeable increase in uneven and rough surfaces. These rough surface areas are probably due to the structure reconstruction of the chemical composition that occurred during the OER process. The SEM images in Appendix A indicate that the reconstructed new species consists of some large-sized nanoparticles in a blocky morphology on the surface of nickel foam. The XRD spectrum in Figure 6b provides clear evidence of material transformation and the emergence of new compounds within MET-Fe/NF throughout the testing. Peaks around 9.2° corresponding to the (1 1 1) plane of MET-Fe suggest that the post-testing sample still retains MET-Fe components. However, the appearance of numerous new peaks in the spectrum indicates the presence of a significant amount of metal hydroxides, including FeOOH (PDF# 22-0353), NiOOH (PDF# 27-0956), and nickel iron-layered double hydroxide (NiFe-LDH, PDF# 49-0188).

XPS spectra further reveal the changes in chemical composition after the test. The full spectrum in Appendix A shows the presence of Ni, Fe, O, N, C, and K elements in the post-test sample, with the newly added K element and a significant increase in the O element resulting from the adsorption of the electrolyte KOH after the test. The Fe 2p spectrum (Figure 6c) with a characteristic peak at 707.9 eV (metal state) shows a significant decrease in this component after the test, indicating that a large proportion of MET-Fe is transformed into other substances. The peak at 711.2 eV corresponding to higher oxidation states of Fe(II) [62,63] indicates the transformation of Fe into iron-based hydroxides during the structure reconstruction process [58,64]. In the C 1s spectrum (Figure 6d), the peak at 289.6 eV [65] corresponding to carbonates in layered double hydroxides confirms the formation of NiFe-LDH during the restructuring of MET-Fe/NF. An increase in the proportion of the C=O peak in the O 1s spectrum (Appendix A) after the test and its shift towards lower binding energy also indicate the transformation of MET-Fe to NiFe-LDH. The FT-IR spectrum of MET-Fe/NF before and after the OER test also indicates the formation of metal oxyhydroxides and metal carbonate hydroxides.

In conclusion, the MET-Fe/NF underwent chemical restructuring after the test, with a significant proportion of the MET-Fe being converted to metal oxyhydroxides or metal carbonate hydroxides such as FeOOH and NiFe-LDH; the presence of NiOOH, with Ni originating from the nickel foam substrate, was also observed. These different metal hydroxides together form a unique hybrid OER catalyst, revealing the material composition responsible for the high OER catalytic activity of MET-Fe/NF.

The high OER activity of MET-Fe-NF originates from the following advantages: (1) iron/nickel (oxy)hydroxides reconstructing from MOFs have been confirmed to be highly efficient OER catalysts; (2) the synergy effect among multiple active components (FeOOH, NiOOH, and NiFe-LDH) enhances the OER performance of the hybrid catalysts; (3) nickel foam (NF) is a good conductive substrate that can considerably accelerate the electron and mass transfer of OER; (4) the high porosity and uniform growth of MET-Fe on the substrate make the reconstructed iron/nickel (oxy)hydroxides highly exposed for OER.

In fact, our work and the literature have gradually shown the poor stability of MOFs in extremely alkaline solutions and their restructuring in metal (oxy)hydroxides; however, this apparent issue might also provide novel synthetic approaches [66]. Lastly, it is important to emphasize that our synthesis technique is easy to use, safe, and economical, and produces catalysts with outstanding activity.

## 4. Conclusions

In this study, we developed the MOF/NF composites using a solvothermal method, directly employing it as a self-supporting electrode for the OER. MOF-coated nickel foam substrates offered superior conductivity and efficient exposure of the solid–liquid–gas interface due to its high porosity. The optimized catalyst MET-Fe/NF demonstrated the best OER performance, achieving a low overpotential of 122 mV at a current density of 10 mA cm^−2^ and exhibiting robust stability over 15 h. Used as the anode in a two-electrode water-splitting setup, MET-Fe/NF||Pt/C/NF enabled continuous hydrogen and oxygen generation at an operating voltage of 1.463 V, highlighting its applicability in practical applications. During the OER process, MET-Fe/NF underwent a chemical transformation to a hybrid catalyst consisting of NiOOH, FeOOH, and NiFe-LDH, as verified by SEM, XRD, and XPS analyses. This transformation introduced a diverse range of active components recognized for their catalytic activity in OER. This study points to the possibility of easily fabricating self-supported MOF-based electrodes for electrocatalytic water splitting. In the future, more advanced in situ instruments should be employed in electrocatalysis to ensure the accurate design of efficient and stable electrocatalysts.

## Figures and Tables

**Figure 1 nanomaterials-14-01168-f001:**
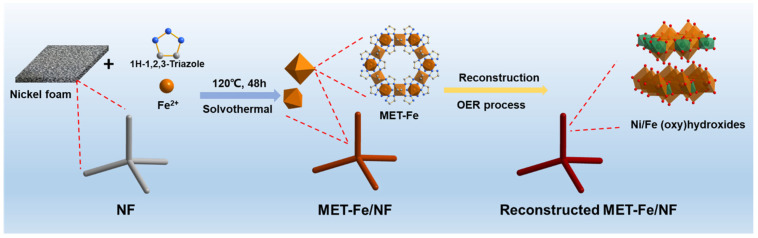
Schematic illustrating the synthesis process and in situ structure reconstruction of the MET-Fe/NF catalyst.

**Figure 2 nanomaterials-14-01168-f002:**
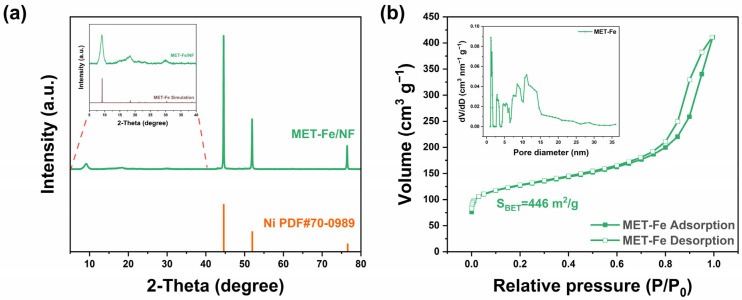
(**a**) XRD pattern of MET-Fe/NF (inset: a detailed view of the dotted area). (**b**) N_2_ adsorption-desorption isotherm of MET-Fe (inset: pore size distribution curve).

**Figure 3 nanomaterials-14-01168-f003:**
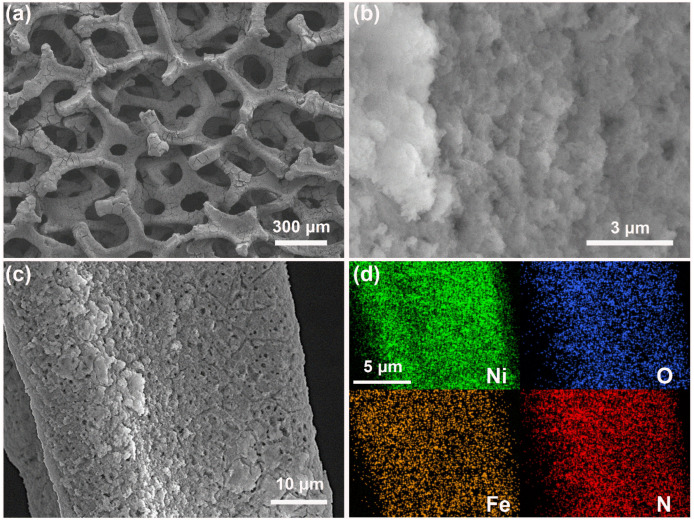
(**a**–**c**) SEM images of MET-Fe/NF at different magnifications. (**d**) Elemental mapping images of MET-Fe/NF.

**Figure 4 nanomaterials-14-01168-f004:**
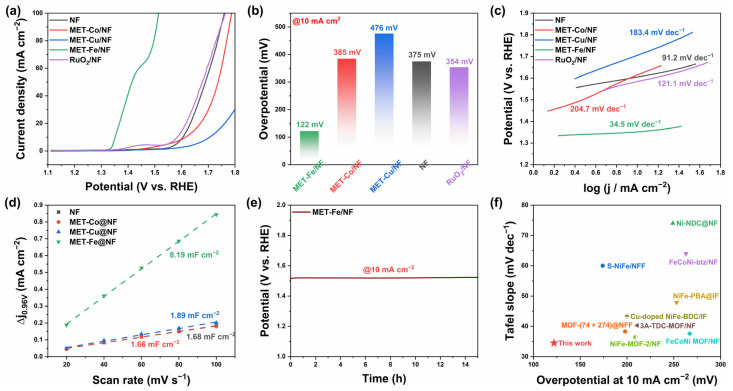
Electrocatalytic OER performance. (**a**) LSV curves. (**b**) Comparison of the overpotentials at 10 mA cm^−2^. (**c**) Tafel plots. (**d**) Electrochemical double-layer capacitance. (**e**) Stability test at constant current density of 10 mA cm^−2^ without iR compensation. (**f**) Comparison of the OER activity of MET-Fe/NF and other reported catalysts: overpotential at 10 mA cm^−2^ and the corresponding Tafel slope.

**Figure 5 nanomaterials-14-01168-f005:**
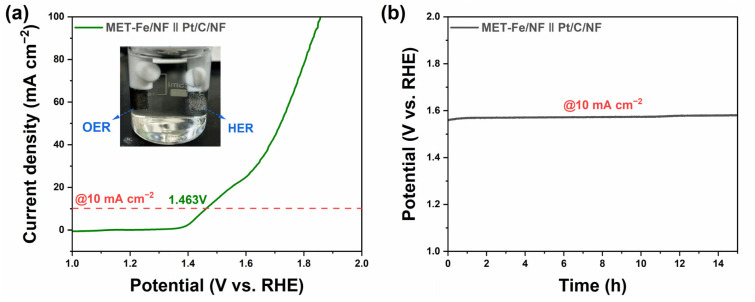
(**a**) LSV curve and photograph of overall water splitting over the MET-Fe/NF||Pt/C/NF two-electrode setup. (**b**) Stability test of overall water splitting at constant current density of 10 mA cm^−2^ without iR compensation.

**Figure 6 nanomaterials-14-01168-f006:**
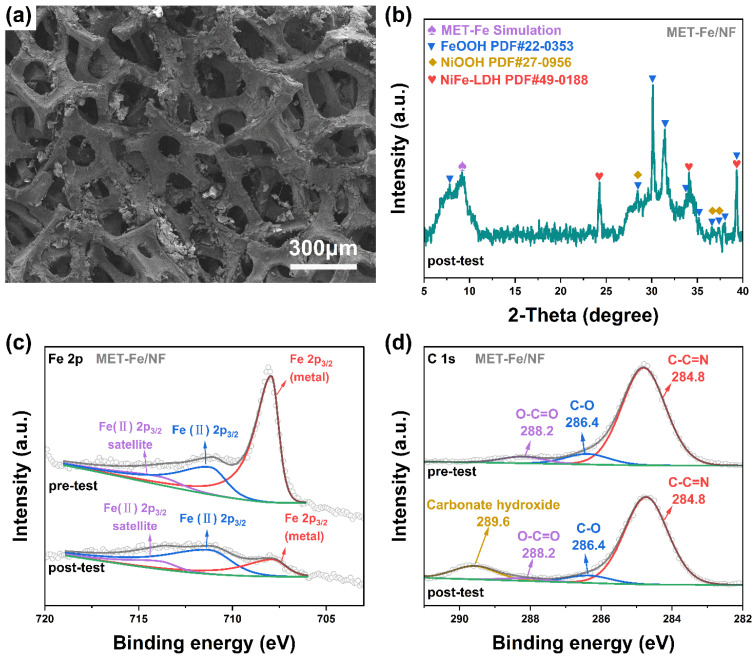
(**a**) SEM images of MET-Fe/NF after OER test. (**b**) XRD pattern of MET-Fe/NF after OER test. High-resolution XPS spectra of (**c**) Fe 2p and (**d**) C 1s of MET-Fe/NF before and after OER test.

## Data Availability

Data is contained within the article or Appendix A.

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
