# Peer review of "Self-Reconstructed Metal–Organic Framework-Based Hybrid Electrocatalysts for Efficient Oxygen Evolution"

_nanomaterials, 2024, doi:10.3390/nano14141168_

Round 1
Reviewer 1 Report (Previous Reviewer 3)
Comments and Suggestions for Authors
1. There are some typographical and grammatical errors that should not be avoided.
2. The OER activity of the MET-Fe-NF is very high. Authors should explain properly. Also, why there is a strong redox peak 1.3-1.4 V?
3. The rational design of the Fe based MOFs based electro catalysts should be explain in the revised manuscript with the reference of following articles.10.1039/D3TA07609A
4. References should be given to formula equations.
5. Purity of chemicals should be mentioned in the revised manuscript.
6. The overall water splitting activity of the MET-Fe/NF should be compared with commercial oxygen catalysts.
7. Can authors explain the electronic structure of the prepared samples with the help of XPS?
Comments on the Quality of English Language
There are some typographical and grammatical errors that should not be avoided.
Author Response
Please see the attachment.

Reviewer 2 Report (New Reviewer)
Comments and Suggestions for Authors
In this manuscript, the authors reported the synthesis of a new type of Metal-Organic Framework (MOF) materials (metal-triazolates) and discussed their self-reconstructed materials forms as electrocatalysts for efficient oxygen evolution reaction (OER).
Of various metal-triazolates, the one based on Fe was found to give the best OER performance, measured in both three-electrode and two-electrode systems, with performances exceeding most of the reported OER catalysts. Overall, this work has high novelty, and the manuscript was well presented. Therefore, this work is worthy to be published in Nanomaterials. Before acceptance, the below comments need to be addressed to further enhance the quality of the manuscript.
1. What are metal-triazolates? Based on the context, it can be inferred they belong to one category of MOFs. However, while MOFs have been largely introduced, not much background information was given to metal-triazolates in the Introduction section. Please provide at least a brief introduction of this type of materials and their unique advantages.
2. To appeal to a broader readership, recent works on MOFs can be included in the Introduction (e.g., SusMat 2021, 1, 460).
3. The authors used Hg/HgO as the reference electrode during their electrochemical measurements. How was this converted to the RHE scale (which was reported in the manuscript)?
4. Recent works on water electrolysis can be referenced in Introduction (e.g., Materials Reports: Energy 2023, 3, 100212).
5. In the Abstract, the authors stated that structural reorganization would result in the formation of numerous iron/nickel hydroxides, however, in the Figure 1 schematic, it was described as Ni/Fe (oxy)hydroxides after reconstruction. Please be consistent in these descriptions.
6. For the electrochemical water splitting in a two-electrode system, not much information was provided in the Experimental, such as the loading of catalysts (for example, how the Pt electrode was prepared and at what loading?).
7. Figure 3d, scale bars for the EDS mapping images should be provided.
Author Response
Please see the attachment.

Reviewer 3 Report (New Reviewer)
Comments and Suggestions for Authors
The manuscript entitled "Self-Reconstructed Metal-Organic Framework-Based Hybrid Electrocatalysts for Efficient Oxygen Evolution" reports a series of metal-tetrazolate MOFs as electrocatalysts for the oxygen evolution reaction (OER). Although some data were provided in the manuscript, a minor revision is required for a better understanding of the current work. Thus, the authors are requested to address the following issues:
1. Stability of MOFs in different chemical environments is challenging. Since a highly alkaline medium was used, the authors should check the chemical stability of all three MOFs in an alkaline medium and discuss this in the results and discussion section.
2. The thermal stability of MOFs and MOFs grown on Ni foam should be provided.
3. FT-IR spectra of MOFs before and after catalysis will help to better understand the formation of multiple metal hydroxides during the catalysis.
4. Authors should discuss in detail why the currently developed MOF-based catalysis outperforms commercial ruthenium dioxide (RuO2).
5. Are these catalysts reusable?
6. A few minor formatting issues should be fixed.
Comments on the Quality of English LanguageMinor editing of English language required
Round 2
Reviewer 1 Report (Previous Reviewer 3)
Comments and Suggestions for Authors
Accept
This manuscript is a resubmission of an earlier submission. The following is a list of the peer review reports and author responses from that submission.
Round 1
Reviewer 1 Report
Comments and Suggestions for Authors
The manuscript describes the preparation and characterization of three composite materials, composed of a microporous MOF coating, supported in a Ni foam (NF). The metal(II) cations in the MOF structure are interlinked by triazolate anions; thus the MOFs are denoted MET-M (M = Fe, Cu, Co). The MET-M/NF composite materials were studied in oxygen-evolution electrocatalysis. The characterization of MET-M/NF is mostly acceptable. The electrocatalysis section, as well as some conclusions, however, need a serious revision. The most notable remarks are provided below.
1) The MOFs are unstable at highly basic conditions (pH = 14), at which the electrocatalytical experiments were carried out. The XRD data (Fig. 7b) clearly show the decomposition of MET-Fe as well as chemical etching of the surface of NF. The lone featureless peak at 2theta ~9 is hardly a reason to claim a preservation of the MOF structure. Therefore, the MET-M/NF is only a precursor to a complex M/Ni-oxo-hydroxo species, which actually participate in the water electrocatalytical splitting. At the same time the authors keep referring it MET-M/NF, as if it is the same original sample, which structure and chemical composition was characterized and established. This is totally misleading. The decomposed samples must be named differently.
2) The whole theoretical section and Fig. 8 are meaningless as there is no more MOF. The proposed mechanism does not have any reasonable basis. Alternatively, the authors must provide unambiguous arguments for preservation of MOF in a reasonable amount to consider its catalytic activity.
3) Complex layered hydroxides of transition metals (LDH) are used in the water electrocatalysis frequently. Could the authors compare the catalytic performance of their samples with some notable literature examples at similar conditions? What is the level of the achievements from the practical point of view? What are possible advantages of the proposed LDH coating preparation via the MOF intermediates over the other, more direct methods?
4) Page 5. How the MET-M samples for the gas adsorption were prepared? Were they somehow detached from the NF support?
5) Figure S3b. Please, scale up the peak intensity al low angles so the MOF crystal structure in the composite could be visualized and identified.
6) Page 9. A reference to Figure 8b is incorrect. There should be Figure 7b (XRD pattern).
Reviewer 2 Report
Comments and Suggestions for Authors
1. What does MET mean ?There is no definition.
2. What is the oxidation state of Fe after the synthesis of MET-Fe ? Normally, Fe is more stable in Fe3+ state. And, how did you identify the oxidation state?
3. This is a major issue I would like to point out. I assume that Figure 8b was calculated assuming that the OER level is 1.23 eV below the RHE .
A. How did you consider the experimental condition of 1M KOH? It should come in through the Nernst equation so that your calculation is consistently based on the computational hydrogen electrode model. Therefore, authors should answer this question by adding one paragraph in the supporting information so that readers can be sure that your procedure is correct.
B. In the DFT calculation, which oxidation state of Fe was used ? For this, authors should refer to comment 2 above.
C. How does the electron correlation effect affect your calculation? Authors can answer this question based on the DFT+U method with a proper U parameter.
Comments on the Quality of English Language
English is very good.
Reviewer 3 Report
Comments and Suggestions for Authors
This manuscript reports on the Self-Reconstructed Metal-Organic Framework-Based Hybrid Electrocatalysts for Efficient Oxygen Evolution. MET-Fe/NF materials was directly grown on Ni-foam through a solvothermal method and used as OER catalyst. This manuscript is well organized. The experimental findings were supported by theoretical studies. However following revisions can be made before publication:
1. Full name should be given to the abbreviations used in the abstract section in their first presence.
2. The prepared MET-Fe, requiring only an overpotential of 122 mV at a current density of 10 mA cm-2. This is exponentially higher performance. Authors should explain significant reason for such a high OER activities.
3. The XRD of pure Ni foam also should be presented.
4. How about the OER activities at higher current densities? The enhanced performance of optimized catalyst should be not only attributed to the increase of the amounts of active sites (is proportionally positive to the value of ECSAs), but also is related to the improved intrinsic activity of active sites. Thus, it would be better to give the results of TOFs. Details of the calculations should be included.
5. I wonder how NF has better performance than MET-CU/NF? Fe 2p fitting is not appropriate.
6. More analysis on the conductivity revealed by DOS should be performed, especially the projected DOS on the mainly involved elements and orbitals.

Extensive editing of English language required
Round 2
Reviewer 2 Report
Comments and Suggestions for Authors
1. What do MET , NF, and LDH stand for ? There is no explanation.
2. Can you write the chemical formula of the MET-Fe? It is not easy to understand Fig. 1(b). Apparently, there seems to be CºN group in Fig. S1(b).
3. Describe in more detail the evidence of metallicity of the MET-Fe, while the MET-Co is insulating. You can draw the electronic DOS around the Fermi level for the two materials.
4. “Figure 5b reveals…” should be “figure 5c reveals….”
5. Figure 8b shows that they considered only single site mechanism (SS).
A. You have to compare the Fig. 8b with the corresponding ones including MET-Co or MET-Cu.
B. There is one more possible mechanism, i.e., dual site mechanism which involves two different sites. So, authors should check if this mechanism is subject to a lower overpotential than that for the SS.
Comments on the Quality of English LanguageEnglish is OK.
Reviewer 3 Report
Comments and Suggestions for Authors
The authors did not respond to most of the comments properly. The reviewer asked for the DOS profile and the authors responded with EIS. Please clarify the relationship between the DOS and EIS in detail.
The reviewer asked for the calculation of the TOFs, and the author responded that TOFs is not necessary for this work.
The authors think that it already meets the standards for publication in the journal.
Round 3
Reviewer 3 Report
Comments and Suggestions for Authors
The authors did not respond to most of the comments properly. The reviewer asked for the DOS profile and the authors responded with EIS. Please clarify the relationship between the DOS and EIS in detail.
The reviewer asked for the calculation of the TOFs, and the author responded that TOFs is not necessary for this work.
The authors think that it already meets the standards for publication in the journal.
Author Response
Thank you for your comments.
We have updated a more proper point-by-point response to your comments on Round 2. However, your comments on Round 3 were just as the same as those on Round 2, which were not the proper and fair comments. Please check our updated response on Round 2. We are looking forward to recieving your kind and constructive comments.